# Algorithms for Vision-Based Quality Control of Circularly Symmetric Components

**DOI:** 10.3390/s23052539

**Published:** 2023-02-24

**Authors:** Paolo Brambilla, Chiara Conese, Davide Maria Fabris, Paolo Chiariotti, Marco Tarabini

**Affiliations:** Department of Mechanical Engineering, Politecnico di Milano, Via La Masa 1, 20156 Milan, Italy

**Keywords:** vision-based quality inspection, defect classification, machine learning, deep learning, image processing, signal processing

## Abstract

Quality inspection in the industrial production field is experiencing a strong technological development that benefits from the combination of vision-based techniques with artificial intelligence algorithms. This paper initially addresses the problem of defect identification for circularly symmetric mechanical components, characterized by the presence of periodic elements. In the specific case of knurled washers, we compare the performances of a standard algorithm for the analysis of grey-scale image with a Deep Learning (DL) approach. The standard algorithm is based on the extraction of pseudo-signals derived from the conversion of the grey scale image of concentric annuli. In the DL approach, the component inspection is shifted from the entire sample to specific areas repeated along the object profile where the defect may occur. The standard algorithm provides better results in terms of accuracy and computational time with respect to the DL approach. Nevertheless, DL reaches accuracy higher than 99% when performance is evaluated targeting the identification of damaged teeth. The possibility of extending the methods and the results to other circularly symmetrical components is analyzed and discussed.

## 1. Introduction

Nowadays, in the Industry4.0 framework, the interest in improving the quality of the products is increasing thanks to the rapid spread of new technologies and the increasing usage of machine learning (ML) and artificial intelligence (AI) approaches at the production level. The application of these new approaches to image-based quality inspection is gaining popularity, as evidenced by the increasing number of research publications in the field of quality control in the last 10 years [1].

An area of interest in several industrial fields is the identification of defects on circularly symmetric components (CSC) such as bearings, gears, buttons, circular saw blades, clutch discs, and washers [2]. These components are produced by automatic plants, and in order to achieve zero-defect manufacturing, it is not possible to adopt the classical visual inspection approach [3]. In the case of CSC, it is possible to perform specific analyses thanks to the knowledge of the symmetry and periodicity of the components.

Since the literature does not address the specific case of defect identification on knurled washers, we decided to perform a systematic literature review following the guidelines for a systematic literature review applied to engineering [4]. Queries were used to search in the academic database; each query included some keywords (“defect detection,” “defect identification,” “computer vision,” “machine vision,” “machine learning,” “deep learning”) bound by a Boolean logic operator. The research resulted in 84 unique titles. Articles were investigated to find information about the field of application, the type of hardware, the acquisition strategies, and the algorithms used.

Among these 84 documents, five are surveys and reviews [3,5,6,7,8]. In the field of application, many works can be traced back to three specific categories: 25 documents are related to the steel industry, 14 to food and agriculture, and 10 to textiles and fabric production.

Different studies aim at identifying superficial defects on non-periodic and non-symmetrical objects such as imperfection in fruits [9], crack in concrete structure [10], dent in metallic surface [7] and scratch on screen [11].

Ten works focused on the identification of defects in CSC; the majority related to the study of bearings [12,13,14,15,16], and some of them [12,13] exploited the transformation of the image from polar to cartesian (P2C) coordinates as proposed in this manuscript. The P2C transformation is used on other products such as the bottom side of bottles [17,18], camera lenses [19], circular polyurethane sealing elements [20], and metal cans [21].

The systematic literature review provided an overview of the data processing approaches that can be used for the generic problem of defect identification. In 18 articles [10,22,23,24,25,26,27,28,29,30,31,32,33,34,35,36,37,38], a deep learning (DL) approach based on neural networks (NN) was used. Some papers exploit custom DL models; many others use known architectures such as ResNet, VGG, U-Net, GoogLeNet, AlexNet, and Yolo. Other works use simple NN as multilayer perceptrons (MLP) [39,40]. For the classification task, it turned out that Support Vector Machine (SVM) is the most used algorithm; it was used in 10 works [19,26,30,33,39,41,42,43,44,45,46]. Apart from SVM, other machine learning (ML) algorithms deserve to be mentioned, e.g., k-Nearest-Neighbors (k-NN) presented in five papers [26,33,45,47,48], clustering algorithms [11,49], and random forest [45]. Out of the DL and ML approaches, many works propose a classification based on decision trees or custom algorithms [17,50,51,52].

Since none of the above listed works focused on washers, the research was extended to include fasteners, washers, and fixtures in the original queries. Existing studies focused on the inspection of fasteners and the detection of missing washers in assemblies [53,54,55,56,57,58,59] and not on the quality of washers and their defects. No dataset including labelled or non-labelled washers with defects with the desired characteristics was found: existing datasets included different washers’ models, images acquired with different optical setups, often with the camera optical axis not coincident with the washer axis.

In this work, we compare the performances of standard algorithms based on feature analysis [2] with DL techniques in the specific application of quality control of knurled washers. The paper is structured as follows: The proposed DL method and the algorithms for data pre-processing are described in Section 2. The case study is presented in Section 3; Section 4 summarizes the experimental results, which are discussed in Section 5, together with the conclusion of our study.

## 2. Materials and Methods

### 2.1. Image Preprocessing

The image pre-processing consists of the identification of the specimen in the image area, the determination of the object center, and the polar to cartesian transformation. These operations are fairly standard and can be applied to any CSC according to the method described in the next few paragraphs. To demonstrate the applicability of this approach to any CSC, we will discuss it considering, as a general test case, a conical gear wheel and a ball bearing, moreover the washer, object of the case study is shown too. Given its transversal applicability, we will still refer to the component as CSC.

#### 2.1.1. Detection of the Specimen

This step allows isolating the image pixels corresponding to the CSC from the background. The optical system must be designed so that the pixels in the background have a different grey scale level from the pixels corresponding to the CSC. In these conditions, the threshold value for the binarization can be automatically selected by the Otsu method [60], which minimizes the weighted variance between the foreground and background pixels. Examples are shown in Figure 1.

#### 2.1.2. Identification of the Object Center

The second step allows identifying the center of the CSC; this passage is crucial since it determines the origin of the polar reference system. The inner and outer edges of the CSC can be detected using the Hough transform [61] for circular primitives, whose result consists of the circles’ centers and diameters.

Results of the application of the Hough transform are shown in Figure 2, which illustrates the outer (green) and inner (red) circumferences and their centers. The center of the CSC $\left(x_{C},\;y_{C}\right)$, reported in blue in Figure 2, is computed as the middle point between the centers of the inner and outer circumferences; this point becomes the origin of the polar reference system.

In this way, it is also possible to evaluate the eccentricity of the sample, which in many circularly symmetric components is considered a defect.

#### 2.1.3. Polar to Cartesian Coordinate Transformation

As evidenced by the literature review, the CSC analysis must be performed by analyzing annuli that are transformed into stripes by the P2C transform (the CSC image is transformed into a polar coordinate system with the origin in $\left(x_{C},\;y_{C}\right)$).

First, a square shaped Region of Interest (ROI) is extracted from the original image; the ROI’s height and width are equal to the diameter of the outer circle, while the center of the square corresponds to the center of the washer. The P2C transformation equation is therefore:(1)$$\left\{\begin{array}{c} x=C_{x}+r\cdot cos\left(\theta \right)\\ y=C_{y}+r\cdot sin\left(\theta \right) \end{array}\right.\;$$
where:
x is the horizontal coordinate of the original cartesian reference system;y is the vertical coordinate of the original cartesian reference system;r is the radius of the polar reference system;θ is the angle of the polar reference system;


The vertical coordinate of the new cartesian reference system is given by the radius itself, while the horizontal coordinate of the new reference system w is computed through Equation (2).
(2)$$\theta =2\pi \left(\frac{w}{W}\right)\;$$
where W is the length of the perimeter of the outer circumference. The result of the P2C transformation is shown in Figure 3. In the picture, the procedure is applied to the gear wheel and to the washer.

### 2.2. Standard Approach

#### 2.2.1. Features Extraction

The image obtained after the P2C transformation is periodic, and it is possible to extract one or more pseudo-signals from the images in the *r*-*w* coordinate system by analyzing groups of rows. Said $\lambda \left(r,w\right)$ the grayscale level of the pixel with coordinates $\left(r,w\right)$, pseudo signals (γ) can be computed as a function of the abscissa w, as the average grayscale levels between belonging to a certain stripe (hereinafter, stripe indicates group of pixels between the rows $r_{0}$ and $r_{M-1}$):(3)$$\gamma \left(w\right)=\frac{1}{M}\overset{r_{M-1}}{\underset{r=r_{0}}{\sum }}\lambda \left(r,w\right)\;\;\;\;\;\;\;\;\;\;\;\;\;\;w=\left[0,W\right]\;$$
where:
γ
is the pseudo-signal describing the grey-scale level along a certain group of rows$\gamma \left(w\right)$
is the value of the wth sample of the pseudo signal, corresponding to a specific horizontal position/angle;*M* is the height of the stripe expressed in pixels;$r_{0}$
is the lower coordinate of the stripe;$r_{M-1}$ is the upper coordinate of the stripe.


An example of pseudo-signal extracted from the conical gear of Figure 3 is shown in Figure 4.

The position and the height of the stripes must be selected depending on the average size of the defect and on its expected location. Good results are usually obtained imposing M between two times and five times the size of the defect expressed in pixels.

The common features that can be extracted from $\gamma \left(w\right)$ are the root mean square (RMS), the standard deviation (SD), the skewness and kurtosis indexes, or other parameters such as the percentiles or the crest factor. An exhaustive list of the possible features that can be extracted from $\gamma \left(w\right)$ is presented in [62,63]

Since $\gamma \left(w\right)$ is a periodic signal, it can also be analyzed in the angle domain using the Fourier transform. The signal Γ can be obtained as a function of the number of repetitions per element (*k*) as follows:(4)$$\Gamma \left(k\right)=\frac{1}{W}\overset{W}{\underset{i=0}{\sum }}\gamma \left(i\right)e^{-j2\pi k\frac{i}{W}}\;$$

The resulting spectrum is shown in Figure 5. In general, the harmonic components that are multiples of the number of periodic elements are clearly visible.

The features that can be extracted from the spectrum are the amplitude of specific harmonic components (for instance, the one indicating the number of periodic components or its multiples) or the spectral centroid [62], the Mean and the Median Angular Frequency [64].

#### 2.2.2. Feature-Based Classification

The input of the classifier is the list of features computed in the previous section, while the output is a discrete index or continuous index expressing the possibility of the presence of defects. As evidenced in the literature review, many algorithms can be used to classify the presence of defects starting from the features extracted from the pseudo signal. The most common ones are probably the SVM, NN, RF, k-NN, and clustering. The choice of the classifier depends on different aspects, such as the complexity and dimension of the input data, the dataset dimension, and the complexity of the classification task.

Many of the algorithms cited above can be tuned with different parameters (the type of kernel in the SVM, the number of neighbors in the k-NN, the number and type of nodes and layers in a NN classifier…). The selection of these parameters must be carried out by maximizing the classification accuracy while minimizing the computational time. In the industrial implementation, the complexity of the algorithms must be limited to allow their execution in the amount of time available for the quality inspection; within the algorithms that can respect this time constraint, the parameters that guarantee higher accuracy are selected.

### 2.3. Deep Learning Approach

The deep learning approach is based on the analysis of a single element of the CSC. The image obtained after the P2C transformation or the stripe (Figure 3 and Figure 4) can be divided into *N* components in order to obtain a representation of a single periodic element, where *N* is the number of repeated elements in the CSC (*N* is 30 in the example of the gear wheel). As an example, Figure 6 shows the division of the image in Figure 3 into 30 parts to obtain 30 images of the periodic element.

The images obtained with this procedure are classified as “compliant” or “defective” using the DL approach. The pictures of the single element (the tooth of the gears) are input tensors of NN models trained for image classification. The classification can be performed using the state-of-the-art architectures; common choices for image classification are MobileNetV2 [65] and ResNet50 [66]. The effectiveness of the DL method in a real industrial application depends on the hyperparameter optimization, which must be chosen as a trade-off between computational time and accuracy. A sensitivity analysis versus the number of epochs, learning rate, and batch size is needed in order to derive the best classification performances in the maximum time available for quality control.

In both cases, we suggest adding data augmentation layers before the main node and the “Dense” layer: given the circular symmetry, data can be easily augmented by adding random rotations and horizontal/vertical shifts. The amounts of rotation and translation must be chosen depending on the size of the image and on the number of periodic components *N*. The obtained images are then normalized, i.e., the pixel values are scaled in range from 0 to 1 by dividing every pixel by the highest intensity value, i.e., 255. A “Softmax” activation is used in the final layer to perform the classification.

## 3. Case Study

### 3.1. Standard Approach

The proposed methods were applied to monitor the quality of knurled washers produced through blanking and coining in an industrial plant. An example of the component is reported in Figure 7. The object is characterized by the presence of 45 teeth placed radially along the whole circumference of the washer. The teeth are the only area of the washer subjected to defects.

The pre-processing steps were applied as described in Section 2 to obtain the cartesian coordinate image.

In the standard approach, the knurled area of the washer was divided into five stripes of equal height. An example of the Cartesian image, a stripe, and the corresponding pseudo-signal are shown in Figure 8. The features extracted from $\gamma \left(w\right)$ were the root mean square value, standard deviation, skewness, and kurtosis.

The Fourier transform $\Gamma \left(k\right)$ of the pseudo-signal is shown in Figure 9, and the number of repeated elements in the CSC is *N* = 45 since the washer has 45 coined teeth. The most relevant harmonics are the multiple of 45.

The features extracted from $\Gamma \left(k\right)$ are the amplitudes of the first eight harmonic components and the mean angular frequency and median angular frequency as described in [64].

Since 14 features are extracted from each of the 5 stripes, each washer is represented by 70 features, which are the input variables of the binary classifier. In this specific case study, we implemented a NN and an SVM. In this implementation, the architecture of the NN is composed of 70 input nodes followed by two hidden layers with eight and four Rectifier Linear Unit (ReLU) nodes, respectively. The last layer of the network is a sigmoid node, so the value returned by the algorithm is between 0 and 1. The number of layers and nodes was manually tuned, selecting the combination granting the highest accuracy; depending on the number of features and on the complexity of the classification, other architectures can lead to better performances [67]. The SVM has been implemented the radial basis function (RBF), the kernel coefficient (gamma parameter) was left to the default value equal to 1/nf where nf is the number of features.

### 3.2. Deep Learning Approach

The teeth of the knurled washer are extracted from the cartesian picture by dividing the cartesian image into 45 parts, as shown in Figure 10.

An example of compliant and defective teeth is shown in Figure 11. The defective samples are sorted from the one with the biggest defect to the one with the smallest defect.

Images were classified using MobileNetV2 and a ResNet50, both completed with a final layer with two “Dense” nodes and “Softmax” activation so that each node represents the probability of the input sample belonging to that class. The data augmentation layer included a random rotation in the range between −5 and +5° and a random horizontal and vertical shift in the interval between −5 and +5 pixels.

The training hyperparameters, such as learning rate, number of epochs, and batch size, were selected after evaluating the classification performances versus different combinations of these parameters. The values selected in the final implementation are shown in Table 1.

### 3.3. Performance Evaluation Procedure

The standard and the deep learning approaches were trained and tested on a dataset (hereinafter the washer’s dataset) composed of 300 washers. In this dataset, 40% of the samples were defective, and the remaining 60% were compliant.

For the standard approach, the dataset was randomly divided into three subsets, each containing 150 pictures for training, 50 for validation, and 100 for testing.

For the deep learning approach, 200 images were split into 45 parts, including 1 tooth each, resulting in 9000 tooth pictures. In most of the cases, the defective washer contains only one or a few defective teeth; the derived dataset is unbalanced and contains 242 defective samples—corresponding to 2.7% of the population. To obtain a balanced dataset, 363 tooth samples were randomly picked from the compliant teeth sample. In this way, a dataset containing 40% defective teeth and 60% compliant teeth was obtained (Figure 12). Further, in this case, algorithm performances were tested on the remaining 100 washers.

The metrics used for the performance validation are reported in Equations (5)–(7):(5)$$accuracy=\frac{\left(TP+TN\right)}{\left(TP+TN+FP+FN\right)}$$
(6)$$Precision=\frac{TP}{\left(TP+FP\right)}$$
(7)$$Recall=\frac{TP}{\left(TP+FN\right)}$$
where:*TP* is the number of true positive classified samples;*TN* is the number of true negative classified samples;*FP* is the number of false positives classified as samples;*FN* is the number of false negative classified samples.

The ground truth was determined by an expert operator of the manufacturing company who actually works as a quality inspector.

## 4. Results

The accuracy, precision, and recall obtained with the standard approach are shown in Table 2; and confusion matrices are reported in Appendix A (Figure A1). The results of the SVM and feature-based NN classifiers are compared.

Table 3 shows the accuracy, precision, and recall obtained by the DL models over the test split of the teeth dataset. The confusion matrices are reported in Appendix A; performances represent the performance of the network in the detection of dental defects.

The comparison of performances between the standard algorithms and the DL ones is not straightforward because of the different datasets (“washer” vs. “teeth”); nevertheless, a washer is defective if at least one tooth is defective. Since the misclassification of a single tooth in most of the cases means the misclassification of the entire washer (it does not happen if two teeth misclassified belong to the same washer) the performances of the DL algorithms are manually recomputed based on a “washer-based” classification. Table 4 shows the results.

The chart in Figure 13 shows the training history of the MobileNetV2 model; on the left, training (blue) and validation (orange) accuracy variation over the training epochs is reported, and on the right, training (blue) and validation (orange) loss evolution is displayed.

Figure 14 shows the evolution of the same variables for the ResNet50 model: on the left, the training (blue) and validation (accuracy); on the right, the training (blue) and validation (orange) loss.

Table 5 shows the result of the classification for some critical cases where the two models classified the specific tooth in different ways:

The computational effort for all the approaches was also evaluated using a personal PC with an Intel Core i7 processor and 16 GB of RAM. Analyses were performed to identify the time requested for pre-processing the image, for the pseudo-signal and tooth extraction, and for the classification of the image (defective/non defective).

The time for the pre-processing routine common for the two approaches is 191.9 ± 9.8 ms (C.I. 95%), computed over the 200 samples of Washers’ training and validation split.The time for the computation of the pseudo signal is lower than 2 ms; a similar value is required to extract the teeth for the DL approaches.The time for the feature-based classification of a washer using the standard algorithm is lower than 1 ms for both the SVM classifier and the NN-based approach.The time for the classification of a tooth is 13.28 ± 0.06 ms (C.I. 95%) for the MobileNetV2 model and 36.71 ± 0.08 ms (C.I. 95%) for the ResNet50, both values were computed on 4500 samples of the teeth test dataset.

The overall time requested for the classification of an image using the standard approach is, on average, 1937 ms and is dominated by the image pre-processing. The average time requested for the classification using DL approaches is between 2075 ms (MobileNetV2) and 2306 ms (ResNet50) in the case of parallelization of the teeth classification and between 789 ms (MobileNetV2) and 1843 ms (ResNet50) in the case of serial classification of the 45 teeth.

Another aspect related to the computational effort is related to the algorithm training. Table 6 reports the number of training epochs for each algorithm, the time required to train one epoch, and the total training time. The models training gets advantages from GPU acceleration.

## 5. Discussion and Conclusions

This paper proposes different methods for the binary classification (damaged/undamaged) of CSC. In the first approach, a custom algorithm is used to extract features from pseudo-signals representing the pixel-wise intensity of image subsets. The second approach consists in the extraction of a ROIs from the image of the specimen; each ROI contains a potentially defective element of the component.

The proposed methods are tested on a case study of a knurled conical washer containing 45 teeth distributed along its circumference; both approaches correctly identify the defective components among the compliant ones.

The performances of the DL approaches on the teeth test dataset evidence the high accuracy and the lower precision and recall. The behavior is due to the strongly unbalanced dataset and the consequent presence of a high number of correctly predicted compliant samples (class 0). The results show that ResNet50 is more sensitive to defects but leads to a higher number of false positives with respect to MobileNetV2. False positive predictions occurred when samples were characterized by the presence of dust that was wrongly identified as a defect. The MobileNetV2 architecture is less sensitive to defects and leads to fewer false positives and more false negatives with respect to the ResNet50.

From the qualitative analysis of the training history of the two DL models, one may assume that the ResNest50 architecture needs a higher number of training epochs to get to a tuned model, while the MobileNetV2 is faster; moreover, the evolution of the accuracy and the loss of the ResNet50 model is subjected to higher variations in consecutive steps. This is a consequence of the fact that the tuning of the training parameters (i.e., learning rate, number of epochs, batch dimension, type of loss function) was carried out manually.

Considering a possible online implementation of the system, the execution time plays a key role in the identification of the optimal strategy. The time required for the classification by the standard algorithm is negligible with respect to the time needed for the image pre-processing (that can be optimized, for instance, by ensuring constant positioning of the system or optimizing the image resolution). On the contrary, as the DL approaches, the time needed for the prediction is not negligible. Even if it is low, it must be considered that it should be executed 45 times for each washer, whereas the standard algorithm requires just one single execution for each washer. This issue can be easily solved by parallelizing the classification on different cores since the classification of a tooth is completely independent from the classification of any other tooth.

The greatest advantage of the DL approach lies in the scalability of the problem. Once the NN is trained, it can be used to classify the specific element independently based on the number of occurrences in the component. For example, once the algorithms are trained for the classification of teeth on the washer, the same models can be used for another washer model with a different number of teeth, given the tooth shape is the same.

## Figures and Tables

**Figure 1 sensors-23-02539-f001:**
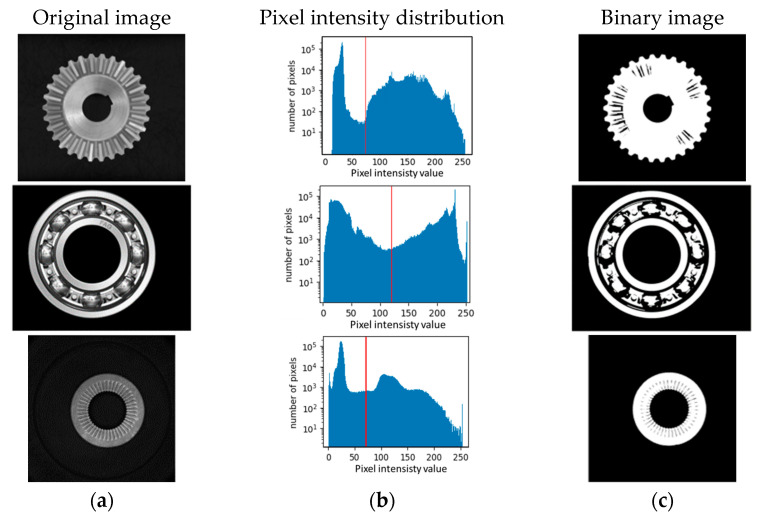
Automatic thresholding of a conic gear wheel (first row) and of a ball bearing (second row) and a washer (third row). Column (**a**) shows the original images; column (**b**) shows the histograms of pixel intensity distribution (the red lines are the thresholds computed with the Otsu method). Column (**c**) shows the binary images.

**Figure 2 sensors-23-02539-f002:**
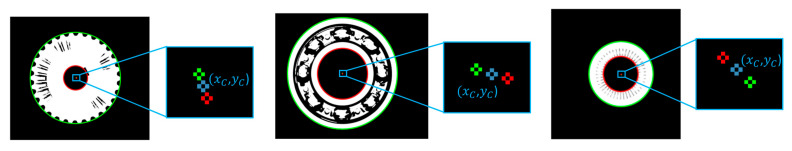
Circumferences found by the Hough transformation on the inner (red) and outer (green) perimeters of the binary picture of the gear wheel, ball bearing and washer. In the right part of each picture the difference in the position of the two centers is highlighted.

**Figure 3 sensors-23-02539-f003:**
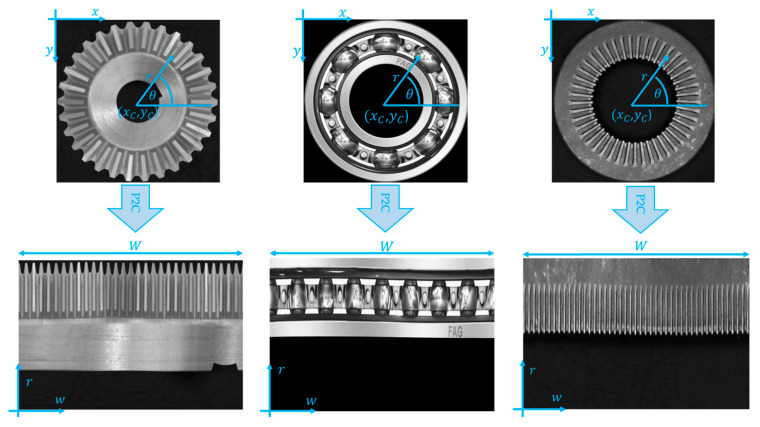
Polar to Cartesian coordinate transformation. The original figures in the Cartesian reference system (*x*, *y*) are shown on the upper row. Their correspondent images in the (*r*, *w*) polar coordinate system are shown on the second row. The transformation is applied on a conical gear, a ball bearing and a washer.

**Figure 4 sensors-23-02539-f004:**
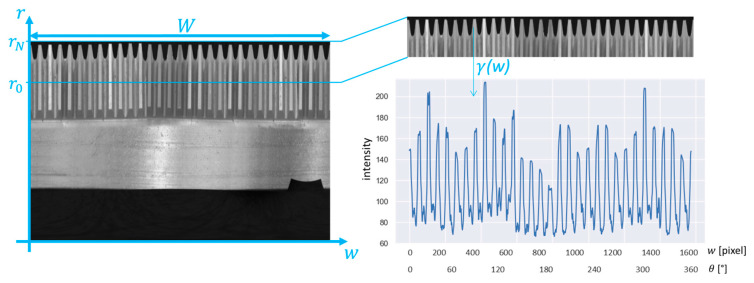
Procedure for the extraction of the stripe from the Cartesian picture and generation of the pseudo signal from the stripe in the case of a gear wheel with 30 teeth.

**Figure 5 sensors-23-02539-f005:**
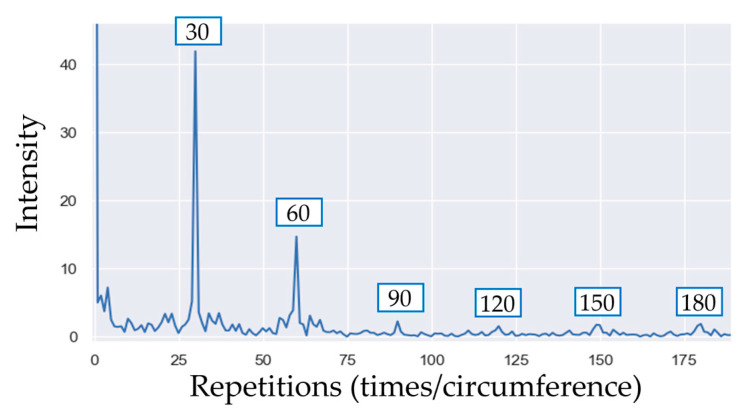
Fourier transform of the pseudo signal shown in Figure 4. The main peak indicates the number of teeth of the CSC (30 in the specific case of the gear wheel).

**Figure 6 sensors-23-02539-f006:**
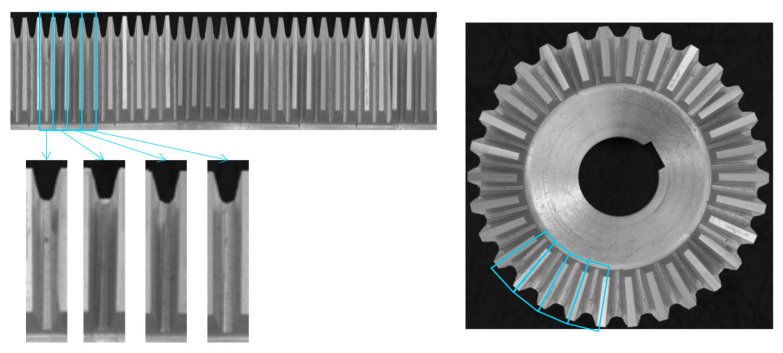
Tooth extraction from the stripe and visualization of the teeth in the original circularly symmetric image of the gear wheel.

**Figure 7 sensors-23-02539-f007:**
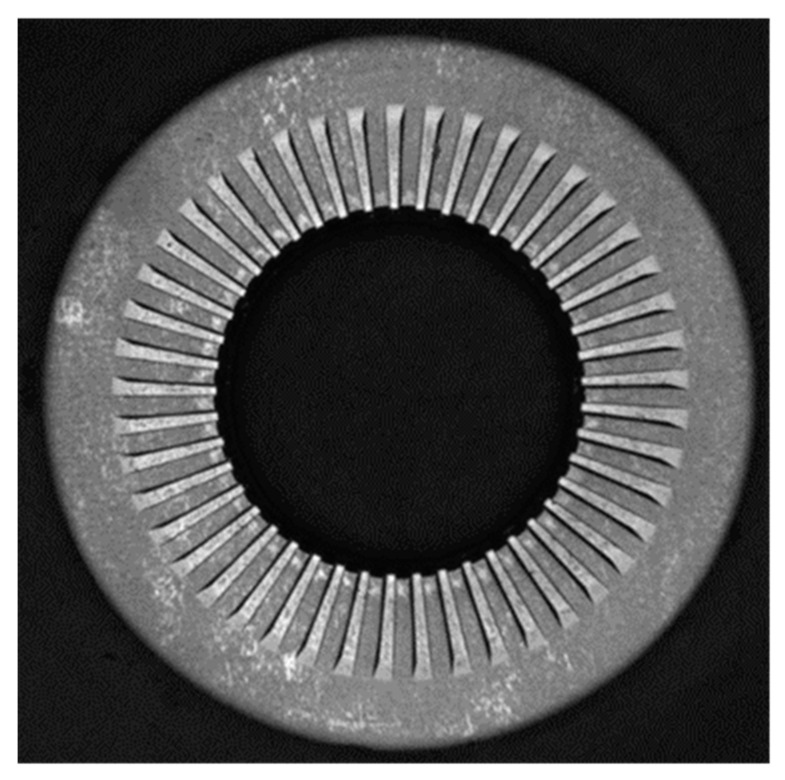
Knurled washer: case study component.

**Figure 8 sensors-23-02539-f008:**
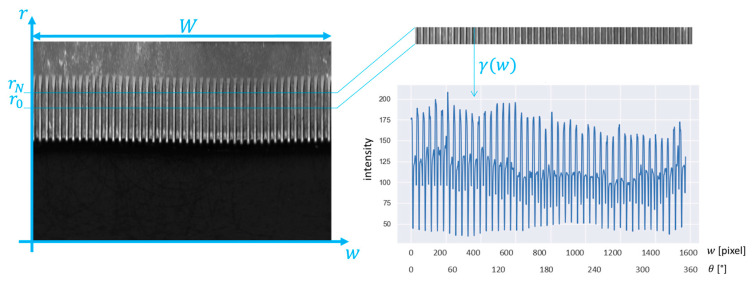
Procedure for the extraction of the stripe from the cartesian picture, and the construction of the pseudo signal from the stripe applied on the knurled washer.

**Figure 9 sensors-23-02539-f009:**
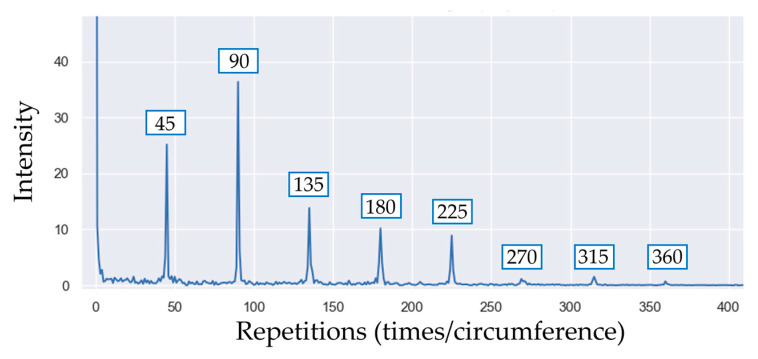
Angular frequency analysis of the pseudo signal shown in Figure 8.

**Figure 10 sensors-23-02539-f010:**
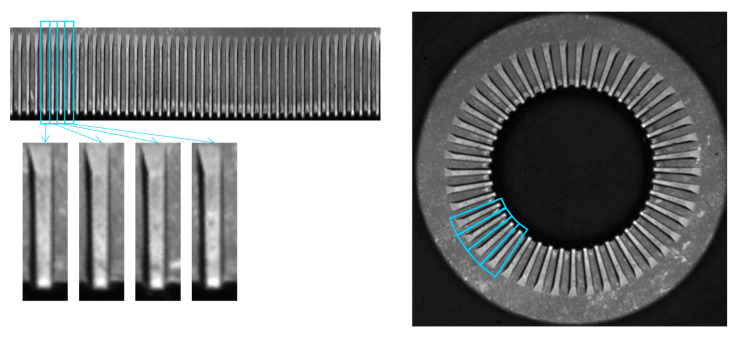
Teeth extraction from the stripe and visualization of the teeth on the original circularly symmetric image of the knurled washer.

**Figure 11 sensors-23-02539-f011:**
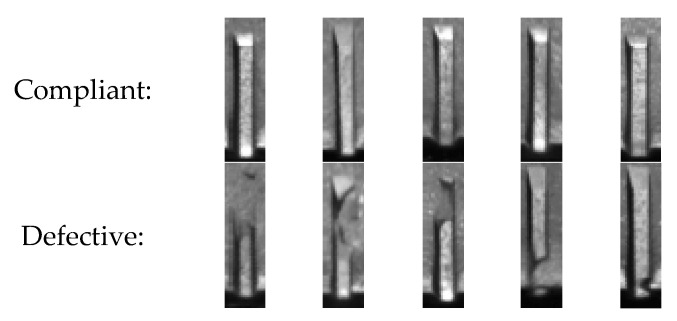
Example of compliant and defective teeth. The defects are sorted from the biggest to the smallest.

**Figure 12 sensors-23-02539-f012:**
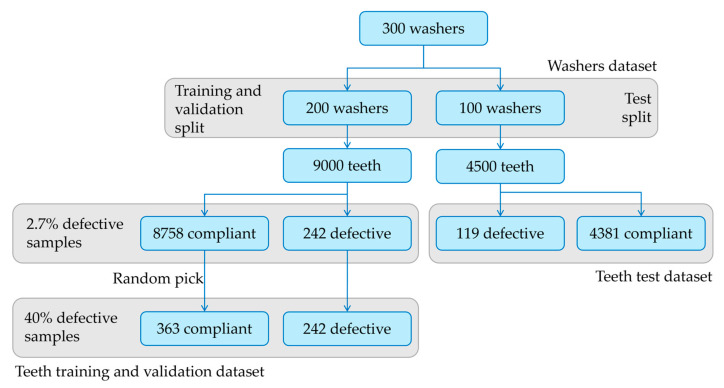
Teeth dataset creation flowchart.

**Figure 13 sensors-23-02539-f013:**
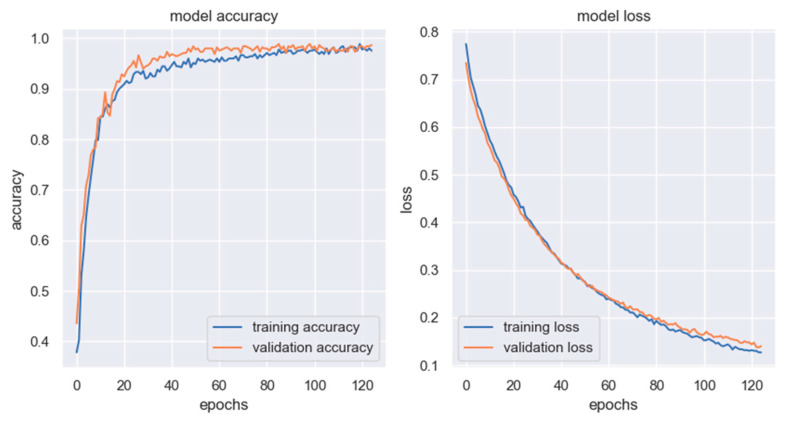
Training history of MobileNetV2 architecture.

**Figure 14 sensors-23-02539-f014:**
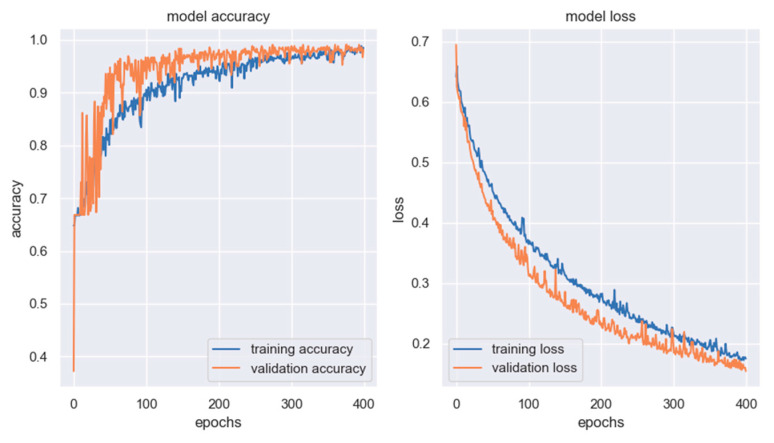
Training history of ResNet50 architecture.

**Table 1 sensors-23-02539-t001:** Deep Learning models hyperparameters.

Model	MobileNetV2	ResNet50
Number of epochs	125	400
Learning rate	0.0001	0.00005
Batch size	32	64

**Table 2 sensors-23-02539-t002:** Classification performances: standard algorithm.

Model	SVM	NN
Accuracy	0.97	0.98
Precision	0.95	0.98
Recall	0.97	0.98

**Table 3 sensors-23-02539-t003:** Classification performances: Deep learning approaches.

Model	MobileNetV2	ResNet50
Accuracy	0.997	0.997
Precision	0.928	0.907
Recall	0.975	0.992

**Table 4 sensors-23-02539-t004:** Performances of the DL algorithms computed based on the washer classification.

Model	MobileNetV2	ResNet50
Accuracy	0.890	0.890
Precision	0.796	0.822
Recall	0.975	0.925

**Table 5 sensors-23-02539-t005:** Misclassified teeth by different models.

	Actual Class	Picture	MoblieNetV2	ResNet50
(a)	Compliant	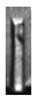	Compliant	Defective
(b)	Compliant	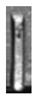	Compliant	Defective
(c)	Defective	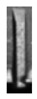	Compliant	Defective
(d)	Defective	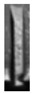	Compliant	Defective

**Table 6 sensors-23-02539-t006:** Training time of the different algorithms.

Algorithm	N° of Epochs	Training Time (One Epoch)	Total Training Time
Standard algorithm—SVM	--	--	<1 s
Standard algorithm—NN	20	<1 s	2 s
DL—MobileNetV2	125	13 s	27 min
DL—ResNet50	400	33 s	220 min

## Data Availability

The data presented in this study are available on request from the corresponding author. The data are not publicly available due to privacy reasons.

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
