# Peer review of "Algorithms for Vision-Based Quality Control of Circularly Symmetric Components"

_sensors, 2023, doi:10.3390/s23052539_

Round 1
Reviewer 1 Report
This study focuses on algorithms for vision-based quality control. A custom signal-based classification is developed to recognize defect. The results of SVM and features-based NN classifier show DL’s performance in dealing with the same problem. The topic is interesting and fits the scope of the journal. However, this article still needs some revisions before its acceptance and the corresponding comments are as given below:
1. In the introduction part, showing the number of documents in each category is too long and useless. In this context, authors are suggested to point out relevant research actuality and significance.
2. The body of the study is not closely related to the circularly symmetric components. It is suggested to add other common defects of circular symmetry problems or change the title of the paper.
3. The part of custom signal-based classification lacks Classification performances such as accuracy, precision and recall. In fact, those are important indicators of the meaninglessness of the algorithm. More related data is needed.
Reviewer 2 Report
The author has done good work on vision-based quality control of circularly symmetric components and the paper is acceptable with minor suggestions as mentioned below.
· The author must provide a few samples of images in dataset section 3.3. All these images the author accessed from a standard database or not?
· The author used only one dataset for authentication of the work. Why author didn’t use 2-3 standard datasets for more authentication of the results?
· The author must provide the mathematical formulation of evaluation parameters in the result section i.e. Accuracy, Precision, Recall, etc. How are they calculated?
· What validation technique was used for finding the accuracy of the proposed work? Justify it.
· The author didn’t mention the hyperparameter value used while validating the deep learning approaches i.e. epochs rate, batch size, learning rate, etc.
Reviewer 3 Report
In this paper, vision technology and artificial intelligence method are combined to solve the quality inspection problem of circular symmetric metric mechanical parts. The classical machine learning method and the deep learning method are used for defect recognition respectively. The former is based on the feature extraction method and the time series analysis strategy, and the latter uses the end-to-end association mapping method. The performance of the two methods in terms of detection accuracy and calculation time is compared through experiments.
The following points need to be addressed:
1. Please explain the purpose of visual inspection of circular symmetrical parts in this paper.
2. In section 2.2, please provide the hyperparameters and structure of SVM and NN model.
3. In section 3.1, how to define the width of the annulus and whether it can provide specific quantitative values
4. In Section 4, it is suggested to supplement the classification confusion matrix.
